# Ultra-Processed Foods and Schooling Are Independently Associated with Lower Iron and Folate Consumption by Pregnant Women Followed in Primary Health Care

**DOI:** 10.3390/ijerph20126063

**Published:** 2023-06-06

**Authors:** Amanda Biete, Vivian S. S. Gonçalves, Sandra P. Crispim, Sylvia C. C. Franceschini, Ariene S. Carmo, Nathalia Pizato

**Affiliations:** 1Graduate Program in Human Nutrition, Department of Nutrition, University of Brasilia, Brasilia 70910-900, Brazil; 2Graduate Program in Public Health, Department of Nutrition, University of Brasilia, Brasilia 70910-900, Brazil; 3Department of Nutrition, Federal University of Paraná, Curitiba 80210-170, Brazil; 4Graduate Program in Nutrition Sciences, Department of Nutrition and Health, Federal University of Viçosa, Viçosa 36570-900, Brazil; 5Ministry of Health, Brasília 70058-900, Brazil

**Keywords:** pregnancy, anemia, processed food, iron, folate, Brazil

## Abstract

Combined deficiencies of nutrients such as iron and folic acid intake during pregnancy are related to nutritional deficiencies risk, such as anemia. The objective of this study was to analyze the association between risk factors (sociodemographic, dietary and lifestyle) and the intake of iron and folate by pregnant women followed up in Primary Health Care (PHC) in the Federal District, Brazil. A cross-sectional observational study was carried out with adult pregnant women of different gestational ages. A semi-structured questionnaire was applied by researchers trained to collect sociodemographic, economic, environmental, and health data. Two nonconsecutive 24-hour recalls (24hr) were carried out to collect data about food consumption. Multivariate linear regression models were used to analyze the association between sociodemographic and dietary risk factors and the consumption of iron and folate. The mean daily energy intake was 1726 kcal (95% CI 1641–1811), with 22.4% (95% CI 20.09–24.66) derived from ultra-processed foods (UPFs). The mean iron and folate intake were 5.28 mg (95% CI 5.09–5.48) and 193.42 µg (95% CI 182.22–204.61), respectively. According to the multivariate model, the highest quintile of ultra-processed foods intake was associated with lower iron (β = −1.15; IC 95%: −1.74; 0.55; *p* < 0.001) and folate intake (β = −63.23; IC 95%: −98.32; −28.15; *p* < 0.001). Pregnant women with high school degree presented higher iron intake (β = 0.74; IC 95%: 0.20; 1.28; *p* = 0.007) and folate intake (β = 38.95; IC 95%: 6.96; 70.95; *p* = 0.017) compared to pregnant women with elementary school degree. Folate consumption was also associated with the second gestational period (β = 39.44; IC 95%: 5.58; 73.30; *p* = 0.023) and pregnancy planning (β = 26.88; IC 95%: 3.58; 50.18; *p* = 0.024). Further research is warranted to enhance evidence on the relationship between the role of processed foods and micronutrients intake to strengthen the nutritional quality of diet of pregnant women attended in Primary Health Care.

## 1. Introduction

Poor nutritional quality of the diet during pregnancy can harm mothers’ health and the development of the fetus. To ensure healthy gestational development, it is important to adequate macro and micronutrient supply, such as carbohydrates, proteins, vitamins, and minerals in order to meet the nutritional needs of both mother and fetus [1,2]. In addition, during pregnancy, it is particularly relevant to consume a variety of fresh and minimally processed foods and water, to meet the need for essential nutrients for this life event, such as iron and folate [3].

The global prevalence of anemia in pregnant women was 36% (95% CI 34–39) in 2019 according to WHO [4]. A recent study identified a prevalence of 23% (95% CI 20–27) in Brazilian pregnant women [5] and those prevalence were considered a moderate-to-severe public health problem [6]. The most common nutritional deficiency during pregnancy is iron deficiency anemia, however, maternal anemia can occur due to several factors, such as acute infections, chronic inflammation, and a single or combined deficiency of nutrients such as folate, and iron [6,7,8]. Iron deficiency anemia occurs due to insufficient iron in the blood [9], and during pregnancy woman’s iron deposits are reduced due to the higher fetus development supply demand [8]. Anemia caused by iron deficiency during pregnancy can have adverse consequences, such as increased premature birth, low birth weight and infant death risk [10,11]. Some factors directly reflect on the predisposition of pregnant women to develop iron deficiency anemia, such as limited schooling, low monthly family income, food insecurity and low consumption of iron-rich foods [12,13,14,15]. In addition to anemia, folate deficiency is linked to neural tube defects in the fetus, intrauterine growth restriction and other fetal malformations, preterm delivery and low birth weight [1,16] and the main risk factors are also related to sociodemographic and dietary factors, such as age, smoking, low socio-economic status, schooling and consumption of folate [17,18].

Considering the importance of adequate food consumption for the prevention of anemia in different publics, it is also necessary to evaluate the nutritional profile and intake of iron and folate during pregnancy, as well as the factors associated with this consumption. Factors such as limited education, low monthly family income and reduced intake of iron-rich foods are associated with the development of iron deficiency anemia during pregnancy [13]. Studies also show the association between education, low income and fewer meals per day with reduced folate intake by pregnant women [13,18,19]. In addition, there is also evidence that the higher consumption of ultra-processed foods (UPFs) by the Brazilian population is related to reduced levels of iron and other nutrients [20].

It is noteworthy that, despite the existence of policies and the Food Guide for the Brazilian Population that recommends the consumption of healthy foods, Brazilian population has replaced the consumption of in natura or minimally processed foods by processed or ultra-processed industrialized foods, foods rich in sugar, fat, sodium and low in fiber and other nutrients [21]. Among pregnant women, different studies presented a high prevalence of UPFs consumption (18.2%–25.4% of daily kcal) [22,23,24]. The UPFs consumption of these foods is associated with greater maternal weight gain, greater neonate adiposity, gestational diabetes mellitus and preeclampsia [23,25,26], as well as the reduction of iron intake and the poor quality of the pregnant woman’s diet [27].

Robust evidence attested by the literature that higher consumption of UPFs is associated with obesity, metabolic syndrome, and cardiovascular diseases in different populations, and to unhealthy outcomes during pregnancy [21,23,25,26]. However, few studies evaluated the association between the consumption of UPFs and iron and folate intake in pregnant women attended in Primary Health Care (PHC). It is also important to evaluate sociodemographic and pregnancy-related factors, such as the gestational period and pregnancy planning, and to dietary habits to better analyze the impact on iron and folate intake. These results may contribute to a better understanding of the determining risk factors associated to these micronutrients intake during pregnancy aiming to support effective intervention actions and to reinforce the Dietary Guidelines for the Brazilian Population. In this context, the objective of this study was to analyze the association between risk factors and the intake of iron and folate by pregnant women followed up in PHC in the Federal District, Brazil.

## 2. Materials and Methods

### 2.1. Study Design and Population

This is a cross-sectional observational study derived from the project “Nutritional status of iodine, sodium and potassium in the Brazilian maternal-infant group: a multicenter study – EMDI Brasil” carried out in the Federal District, Brazil, between August 2019 and September 2021.

To perform this study a simple random sample was calculated using the StatCalc tool of the EpiInfo Software version 7.2 (Center for Disease Control and Prevention, Atlanta, GA, USA), considering the average monthly prenatal care appointments at PHC in 2016 as a proxy for the number of pregnant women monitored by PHC in Federal District (*n*= 18,877—data reported by the Federal District State Department of Health). The prevalence of the indicator “consumption of ultra-processed foods the day before” among Brazilian pregnant women monitored by PHC (81.5%), in the same year, from the Food and Nutrition Surveillance System – SISVAN [28]. The acceptable error of 5% and the 95% confidence interval (95% CI) were considered. At the end, the minimum number of pregnant women to be included was defined as 190, and to anticipate possible sample losses, 10% was added to the estimated number. Thus, the sample was estimated at 209 pregnant women. Also, ten PHC units were selected according to the proximity to the central region and the highest monthly average prenatal care performed in 2016 (data reported by the Federal District State Department of Health), resulting in a representative sample of the studied population.

### 2.2. Inclusion and Exclusion Criteria

Pregnant women from all gestational age (first, second and third trimesters) who attended the selected PHC units for prenatal follow-up appointments were invited to participate at each unit. Pregnant women who were not able to answer the questionnaire or decline were excluded from the sample. Pregnant who had any physical changes that could distort the research results, such as a history of thyroid disease or surgery, a reported diagnosis of hypo or hyperthyroidism, were excluded from the study, as they could interfere with the primary outcomes of the EMDI multicenter study. Figure 1 details the eligibility process.

### 2.3. Data Collect

Data collection was conducted from August 2019 to September 2021, through face-to-face interview. A semi-structured questionnaire was applied by researchers trained to collect sociodemographic, economic, environmental and health data from pregnant women as follow: maternal age (in years), access to government social program (yes or no); self-reported skin color (white, brown/black, or asian); maternal education (no education, elementary school, high school, or higher education); household income in the previous month (up to USD 94.00; between USD 94.00 and USD 188.00; between USD 188.00 and USD 566.00; above USD 566.00 or not informed); use of iron supplementation (yes or no); use of folic acid supplementation (yes or no); use of multivitamins (yes or no); home-made meal frequency (in days); previous pregnancies (none, between 1 and 3, between 4 and 6, or more than 6) and planned pregnancy (yes or no). Information regarding anthropometry (pre-gestational weight, current weight, height), and obstetric history (date of last menstrual period, date of last ultrasound) were collected from the Pregnant Woman’s Handbook. Nutritional status was classified according to WHO pre-pregnancy BMI values (<18.5 underweight, ≥18.5–24.9 normal weight, ≥25.0–29.9 overweight and ≥30.0 obese) [29].

The first 24-hour recall (24hr) was carried out to collect data on food consumption. A second 24hr was collected randomly by telephone in 20% of the sample up to 15 days after the first on non-consecutive days to correct the within-person variability [30,31].

During the interview to apply the 24hr, the US Department of Agriculture’s 5-step multiple-pass method was used, according to the 5 steps established by the institution: (1) initial quick list of foods and beverages consumed; (2) list of commonly forgotten foods; (3) investigation of meal times and occasions; (4) detailing of previous information including description of quantity, method of preparation, additions and brands; and (5) final review of recall information [32]. To assist pregnant women in quantifying the portions consumed, the Photographic Manual of Food Quantification was used, which contains photos of portions, forms of food and household measures [33]. 

The food consumption data from 24hr were entered for nutrients composition using the Globodiet software, Brazilian version, Data Entry mode, developed by the GloboDiet Initiative [34] and analyzed for nutrient composition using the Brazilian Food Composition Table (TBCA) [35]. Analysis of food consumption included assessment of total energy intake, calories consumed from UPFs, and iron and folate intake. The macronutrients were expressed as the percentage of energy intake (%TEI) and the nutritional density of each micronutrient in the diet was expressed in mg or µg per 1.000 kcal [34] while an assessment of nutrient and food intake was categorized according to the processing degree as defined by the NOVA classification, which consists of dividing foods into four groups: in natura or minimally processed, culinary ingredients, processed and ultra-processed [36,37]. For the present study, the percentage of relative energy intake from UPFs were distributed into quintiles according to the contribution of UPFs to the total caloric value of the diet (% kcal), according to the cut-off points of the distribution quintiles. The assessment of iron, folate and energy consumption by pregnant women was also performed using the 24hr. Nutrients from food supplementation were not considered in the calculation of food consumption.

### 2.4. Data Analysis

Descriptive analysis was performed, with calculation of relative frequency distributions for categorical variables and mean and 95% confidence interval (95% CI) for continuous variables. Bivariate analysis was performed using simple linear regression considering the consumption of iron and folate (mg/1000 kcal) as the outcome variable. Multiple linear regression models were performed to predict the consumption of iron and folate (mg/1000 kcal), based on explanatory variables such as age, gestational age (categorized in trimester), family allowance (no/yes), schooling (categorized into 4 categories), meal at home (less than 3 days a week/more than 3 days a week), pre-gestational BMI (categorized), planned pregnancy (no/yes) and consumption of ultra-processed foods (categorized into quintile of total caloric value). It should be emphasized that the minimum sample size for conducting multiple linear regression is 146, considering that this study is evaluating ten independent variables and two adjusting variables [38].

For the construction of the multiple linear models, the *p* ≤ 0.20 value obtained in the bivariate analysis was used as criteria for the inclusion of explanatory variables. In the final models, the Backward Method was used, and those variables with lower significance (higher p-value) were removed one by one. The procedure was repeated until all variables present in the models had statistical significance (*p* < 0.05). The models were adjusted by the variables “ferrous sulfate supplementation” or “folic acid supplementation” (no/yes) and “use multivitamin” (no/yes). The significance of the final models were evaluated by the F test of the analysis of variance and the goodness of fit by the coefficient of determination (R^2^). Residuals were evaluated according to normality, homoscedasticity, linearity and independence assumptions. In addition, multicollinearity was verified between the variables included in the models.

All analyzes were performed using the Stata statistical program, version 17.0 (StataCorp. College Station, TX, USA).

### 2.5. Ethical Aspects

The study was submitted and approved by the Research Ethics Committee of the Faculty of Health of the University of Brasília (UnB), under number 2,977,035 and by the Research Ethics Committee of the Federal District Health Department under number 3,489,243. All pregnant women signed the TCLE before starting the interview.

## 3. Results

The study included 217 pregnant women (Figure 1) aged between 20 and 50 years. It was observed that 44.65% of the pregnant women were in the third gestational trimester. Regarding sociodemographic information, 75.6% of pregnant women declared themselves to be brown or black and 11.74% were beneficiaries of social welfare income transfer program. About schooling, most (52.8%) completed high school degree. Regarding pre-gestational nutritional status, 46.2% were overweight or obese. Also, 66.20% were multiparous, and 62.8% reported not having planned the pregnancy (Table 1).

**Figure 1 ijerph-20-06063-f001:**
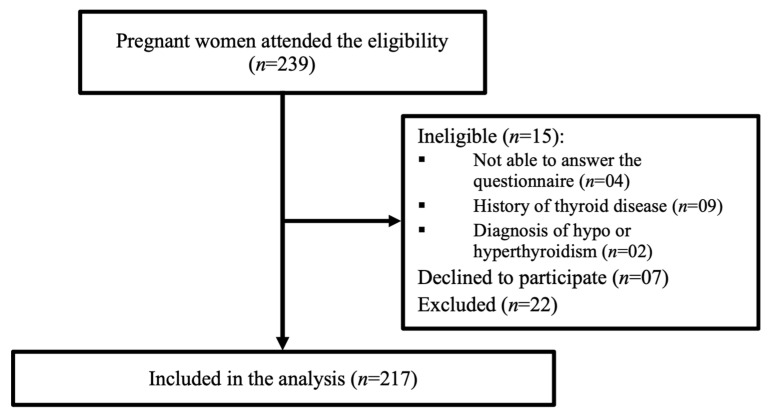
Eligibility screening process sample flowchart. A total of 217 pregnant women were enrolled to the study.

The daily energy intake was 1726 kcal (95% CI 1641–1811). Carbohydrates contributed 50% and protein 17% of TEI, and total fat contributed 33%. Considering the NOVA classification of food processing degree, the participants consumed 64.6% (95% CI 62.18–66.95) of total energy from unprocessed or minimally processed food, 4.6% (95% CI 4.00–5.16) from culinary ingredients, 8.5% (95% CI 7.11–9.81) from processed food and 22.4% (95% CI 20.09–24.66) from UPFs. The mean iron intake was 5.28 mg (95% CI 5.09–5.48) and folate intakes was 193.42 µg (95% CI 182.22–204.61). 

Table 2 and Table 3 present the variables associated with the consumption of iron (mg/1000 kcal/day) and folate (µg /1000 kcal/day) by pregnant women.

Table 4 describes the final model of multiple linear regression of variables associated with iron consumption during pregnancy. The independently associated factors with iron consumption were schooling and UPFs intake. The highest quintile of consumption of UPFs was associated with lower consumption of iron. Pregnant women with high school education degree presented higher iron intake compared to those with elementary school degree. According to the standardized beta assessment, the factor that most influenced iron consumption was the consumption of UPFs.

Table 5 describes the final multiple linear regression model of variables associated with folate consumption during pregnancy. The factors independently associated with folic acid consumption were schooling, gestational age, planned pregnancy, and consumption of UPFs. Pregnant women with high school education degree showed higher consumption compared to those with elementary school degree, the second trimester participants presented higher consumption of folate compared to those who were in the first trimester, and those who had a planned pregnancy also showed higher consumption of folate. The highest quintile of UPFs consumption was associated with lower consumption of folic acid. According to the standardized beta assessment, the factor that most influenced the consumption of folate was the consumption of UPFs.

## 4. Discussion

The presented data, according to the multivariate model, highlighted the association of higher UPFs consumption and low iron and folate intake in pregnant women assisted by PHC. In addition, high school degree, second gestational period and pregnancy planning were associated with a higher iron and folate intake. This is the first study to evaluate the impact of UPFs on iron and folate micronutrient content of the diet consumed by pregnant women in the capital of Brazil. The scientific community has recently focused attention on the importance of adequate nutrition during the first one thousand days of life, period from pregnancy to early childhood. The deficiency of iron and folate nutrients are related to nutritional deficiencies, anemia, and can lead to serious outcomes, such as an increased risk of infant death, such as increased birth weight and infant death [10,11].

The mean iron intake and folate of the evaluated pregnant women were below the requirement during pregnancy [39]. It is well known that the two major sources of dietetic iron are animal foods (beef, liver, offal, poultry, pork, fish and shellfish) and plant foods (dark green vegetables, beans, lentils) [40,41]. Folate is present in dark green vegetables as well as in fortified foods (in the form of folic acid) [42,43]. The pregnant women should be vigilant about their food intake to maintain the adequate levels of micronutrients. To reinforce the adequate consumption of *in natura* and minimally processed foods, sources of iron and folate according to Dietary Guidelines for the Brazilian Population, seems to be an important strategy to prevent micronutrients deficiency. Although, it has been observed due to the physiological requirements, that pregnant woman group will hardly be able to obtain the adequate amount of iron only through food intake, which justifies the need for supplementation at this stage [39,44]. 

A systematic review with studies from Ethiopia, Kenya, Nigeria and South Africa showed higher prevalence of anemia, iron deficiency, and iron deficiency anemia in pregnant women when compared to women of reproductive age, but the folate inadequacy was reduced in pregnant women due to micronutrient fortification and supplementation [45]. The ECLIPSES study, a longitudinal study conducted on 793 non-supplemented pregnant women in Spain, showed that women did not increase their energy and nutritional intake during pregnancy and postpartum and they had a high risk of iron and folate deficient intake [46]. Others studies in French-Canadian [47], Nordic population [48] and US [49] showed iron and acid folic inadequacy in pregnant women. Studies carried out in different Brazilian states showed a higher prevalence of insufficient iron and folate intake by pregnant women. In Minas Gerais State, a study with 492 pregnant women observed a prevalence of 49.2% of insufficient folate intake considering the diet with fortified foods [18] and in Rio de Janeiro State, a prevalence of 63.7% of folate deficiency was reported by 201 pregnant women who entered the study [50]. Other study carried out in São Paulo State with 30 pregnant women, also showed inadequate consumption of iron and folate [19]. A Cochrane review’s meta-analyses evaluating 21 clinical trials (involving 5490 pregnant women) showed that the intermittent oral iron and folic acid supplementation may be a feasible alternative to daily iron supplementation among pregnant women populations where the prevalence of anemia is less than 20% [51]. These results taken together highlights the importance of dietary diversification, food fortification and drug supplementation as a combined strategy to assure iron and folate adequacy to the pregnant group.

In Brazil, two health public policies help to control iron deficiency anemia in pregnant women and children. First, the mandatory fortification of wheat and corn flours with iron and folic acid by food industry, which consists of the mandatory addition of 4 to 9 mg of iron and 140 to 220 μg of folic acid in wheat and corn flour per 100 g of the product [52]. Several countries in South and Central America have also instituted food fortification as a combat resource nutritional deficiency. Costa Rica, Chile, El Salvador, Guatemala, Honduras, Mexico, Nicaragua, Panama, Puerto Rico have institutionalized fortification policies [53]. Fujimori et al (2011) evaluated the retrospective data from 12,119 medical records of Brazilian pregnant women and showed a reduction in anemia prevalence (fell from 25% to 20%) after flour fortification with iron [54]. 

Second, the National Iron Supplementation Program (NISP), which recommends prophylactic supplementation of 40 mg of elemental iron daily throughout pregnancy and 120 mg in the case of already anemic pregnant women [15,16,41,55], whereas micronutrient supplementation are available for free to pregnant women assisted by PHC. Although the program is national, the resource is decentralized, and each municipality is responsible for purchasing and distributing the supplement. About one third of the PHC participants declare to not use iron supplementation, despite the gratuity of the program, this gap in the coverage of the NISP may be related to factors such as the unavailability of the supplement in the PHC units, the unavailability of medication or distribution without proper guidance, the non-attendance of pregnant women in medical appointments or their non-adherence to the prescription [12,56]. 

The total caloric value mean from ultra-processed foods by pregnant women in our study was 22.4%, and the highest quintile of consumption of UPFs was associated with lower consumption of iron and folate. The study of Graciliano et al (2021) evaluated a northeast region Brazilian pregnant women cohort and observed similar UPFs consumption (22.2%). Also, showed an association with reduced consumption of rice, beans, meat, fruits and vegetables, sources of iron and folate [27]. Ultra-processed foods are commonly high in sugar, fat, sodium, with attractive sensory characteristics, not stimulating satiety because they are low in fiber and generally obesogenic. Although most of the UPFs in Brazil contain synthetic micronutrients, the bioavailability of these nutrients is not known and, therefore, it cannot be said that they will have the same bioavailability as in natura or minimally processed foods [36,37]. 

Consumption of ultra-processed foods is associated with greater maternal weight gain and greater adiposity in the newborn [23,25], and with the reduction of iron intake and the quality of the pregnant woman’s diet attended by PHC in Brazil [27]. Another cross-sectional study carried out in Brazil with 125 high-risk pregnant women in the third trimester of gestation showed the majority of the calories intake was from unprocessed foods (52.4%), followed by UPFs (25.5%). The fiber, calcium, folate and iron consumption was below the recommendations for pregnant women, corroborating our results [22]. 

The study of Crivellenti et al., aimed to develop a Diet Quality Index Adapted for Pregnant Women and showed a low ratio of women who reached the maximum score for the consumption of folate and iron, suggesting poor quality of the diet in pregnant women [57]. Consumption of UPFs was the main predictor of iron and folate intake. This result deserves to be highlighted given the importance of these nutrient for the prevention of anemia, a condition that can increase maternal and child morbidity and mortality. According to Nilson et al. (2022) approximately 57,000 premature deaths were estimated as attributable to the UPF consumption in Brazil in 2019 [58], reinforcing the impact of industrial food processing on preventable deaths. Thus, considering the increasing consumption of ultra-processed foods in the Brazilian population in general and in pregnant women, the high consumption of UPFs during pregnancy has important clinical significance given its negative impactive on maternal and neonatal health [22,23,24,59]. Interestingly, despite the influence of dietary energy density on the nutritional profile of the whole diet, there is no specific value for UPFs intake during pregnancy. The consumption of UPFs impacts the quality of the pregnant woman’s diet and it is necessary to encourage the consumption of fresh and minimally processed foods at the expense of UPFs consumption by pregnant women, as recommended by the Dietary Guidelines for the Brazilian Population [37].

The pregnant women with high school degree education had higher iron and folate consumption compared to those with elementary school. Others studies conducted in Brazil also found association between low education levels, insufficient folate intakes in pregnant women and iron deficiency anemia [13,18,19]. Rodrigues et al (2015) showed an association between low folate consumption and low income, in addition to a higher prevalence in younger pregnant women and in those who had fewer meals per day [18]. Regarding the quality of the diet of pregnant women, a study carried out in China showed that limited schooling, occupation and low income were also negatively associated with adequate diet [60]. According to the United Nations Development Program - UNDP [61], the mother’s education level influences on adequate prenatal care during pregnancy. Indeed, women with higher education degrees and higher-income have significantly higher chances of having at least six prenatal medical appointments and one puerperal appointment than those in a more unfavorable situation [62].

Prenatal care adherence during pregnancy is an important health factor associated with optimum outcomes. It is possible to monitor the baby’s development and maternal health, directly impacting the mother’s and child’s health. Study shows that the late initiation of prenatal care is associated with the development of anemia during pregnancy and deserves greater attention from the public sector [12]. Another study showed a higher prevalence of anemia among women who did not undergo adequate prenatal care and pregnant women who had multiple pregnancies, showing the importance of adequate health care as a factor in the prevention of anemia and adverse gestational outcomes [63].

The pregnancy planning was associated with a higher consumption of folate. This result corroborates a study carried out in Minas Gerais State, which also associates a higher prevalence of inadequate folate intake in women who did not plan the pregnancy [18]. Our hypothesis is that pregnant women who had planned their pregnancies are more careful with the nutritional profile of diet and, therefore, have a higher consumption of folate for food. Our study did not consider folic acid supplementation in food intake analysis; therefore, our results cannot extend the supplementation interpretation. Considering that changes in quantitative and/or qualitative nutrient intake may play a crucial role in fetal programming and consequently influence the presence of non-communicable disease in life, it is worth to discuss pregnancy planning as an important tool to enhanced nutritional profile of diet and to enhance adherence to the folic acid supplement [64,65]. 

Regarding the gestational period, the second trimester participants presented higher consumption of folate compared to those who were in the first trimester. A study carried out with 79 pregnant women in Canada evaluated changes in diet quality according to the gestational trimester using the Canadian Healthy Eating Index (HEI). Despite not finding variation in the total score, the authors showed that the adequacy of the diet, fruit consumption and total vegetables, saturated and unsaturated fats decreased significantly throughout pregnancy [66]. This finding suggests the importance of the specialized multidisciplinary care throughout pregnancy.

Possible limitations of this study are related to the cross-sectional study design that does not allow causality analysis. The self-reported 24hr assessment depends on participant memory, cooperation, and communication abilities, and might contain some errors due to memory biases and over/underreporting. Lastly, the consumption of supplements in the food intake analysis was not considered, since there was not enough data collected. Despite these limitations, it is important to highlight the study strengths. To date, this is the first study evaluating iron and folate intake with data from Brazilian pregnant women from the Federal District region with robust methods. The dietary assessment was applied by well-trained nutrition researchers using the 5-steps multiple pass methodology. In addition, to enhance accuracy of dietary recall and to reduce bias the photographic manual of food portions was used when collecting food intake data. The results of the present study indicate important public health implications, since the higher UPFs consumption was associated with lower intake of iron and folic acid nutrients, leading to a poor nutritional profile of the diet of pregnant women assisted by PHC. 

## 5. Conclusions

The consumption of ultra-processed foods and schooling were the main factors independently associated with both iron intake and folate intake in pregnant women attended in PHC. Folate consumption was also associated with the gestational period and pregnancy planning. Therefore, we suggest reinforcing the recommendations of the Dietary Guidelines for the Brazilian Population, to focus on the importance of pregnancy planning, and to reinforce the importance of early prenatal care in order to reduce adverse maternal and neonatal outcomes. 

## Figures and Tables

**Table 1 ijerph-20-06063-t001:** Characteristics of pregnant women assisted in Primary Health Care. Federal District, Brazil, 2019–2021 (n = 217).

Variables	n	%	95% CI
Age			
20 to 24 years	71	32.7	26.77; 39.27
25 to 29 years	55	25.3	19.97; 31.59
30 to 34 years	53	24.4	19.13; 30.61
≥35 years	38	17.5	12.98; 23.18
Gestational Period			
1st Trimester	33	15.3	11.10; 20.84
2nd Trimester	87	40.0	33.62; 46.72
3rd Trimester	97	44.7	38.10; 51.39
Access to social welfare program			
No	188	88.3	83.17; 91.96
Yes	25	11.7	8.03; 16.82
Schooling			
Elementary School	33	15.4	11.15; 20.93
High school	113	52.8	46.06; 59.43
University education	68	31.8	25.85; 38.35
Self-reported skin color			
White	44	20.7	15.71; 26.65
black/brown	161	75.6	69.33; 80.91
Asian	8	3.7	1.81; 7.35
Household Income (USD)			
Up to 94.00	12	5.6	3.21; 9.68
From 94.00 to 188.00	17	8.0	5.00; 12.49
From 188.00 to 566.00	96	45.1	38.47; 51.83
Over 566.00	59	27.7	22.07; 34.12
Not reported	29	13.6	9.60; 18.94
Ferrous sulfate supplementation			
No	115	54.0	31.27; 44.28
Yes	98	46.0	55.71; 68.72
Folic acid supplementation			
No	115	54.0	47.22; 60.61
Yes	98	46.0	39.38; 52.77
Use of multivitamins			
No	180	85.3	79.82; 89.49
Yes	31	14.7	10.50; 20.17
Home-made meal frequency			
Less than 3 days	28	13.1	9.21; 18.41
Greater than or equal to 3 days	185	86.9	8.15; 9.07
Pre-gestational BMI ^1^			
Underweight	98	49.8	42.77; 56.73
Eutrophy	8	4.1	2.03; 7.94
Overweight	58	29.4	23.46; 36.22
Obesity	33	16.7	12.13; 22.66
Previous Pregnancy			
Primiparous	72	33.8	27.73; 40.45
Multiparous	141	66.2	59.54; 72.26
Planned Pregnancy			
No	135	62.8	56.09; 69.02
Yes	80	37.2	30.97; 43.90

^1^ BMI: Body Mass Index.

**Table 2 ijerph-20-06063-t002:** Simple linear regression models of variables associated with iron consumption (mg/1000 kcal/day) of pregnant women assisted in Primary Health Care. Federal District, Brazil, 2019–2021 (n = 217).

Explanatory Variables	Mean Iron Consumption (mg/1000 kcal/day)	CI 95%	Non-Standard β	CI 95%	*R-Squared*	*p*-Value
Age					0.0184	
20 to 24 years	5.30	5.00; 5.60	(reference)			
25 to 29 years	5.07	4.68; 5.46	−0.23	−0.74; 0.28		0.374
30 to 34 years	5.59	5.08; 6.10	0.29	−0.23; 0.81		0.277
≥35 years	5.14	4.79; 5.49	−0.16	−0.74; 0.41		0.576
Gestational Period					0.0050	
1st Trimester	5.43	4.70; 6.16	(reference)			
2nd Trimester	5.35	5.06; 5.63	−0.84	−0.68; 0.51		0.782
3rd Trimester	5.17	4.90; 5.44	−0.26	−0.85; 0.33		0.388
Access to social welfare program					0.0303	
No	5.37	5.16; 5.58	(reference)			
Yes	4.58	3.97; 5.19	−0.79	−1.40; −0.17		0.012 *
Schooling					0.0319	
Elementary School	4.67	4.19; 5.15	(reference)			
High school	5.40	5.15; 5.65	0.73	0.16; 1.30		0.012 *
University education	5.37	4.97; 5.77	0.69	0.08; 1.31		0.026 *
Self-reported skin color					0.0026	
White	5.34	4.91; 5.77	(reference)			
black/brown	5.28	5.05; 5.52	−0.60	−0.56; 0.44		0.813
Asian	4.93	4.31;5.55	−0.41	−1.52; 0.69		0.465
Ferrous sulfate supplementation					0.0046	
No	5.41	5.04; 5,77	(reference)			
Yes	5.20	4.97; 5.43	−0.20	−0.61; 0.20		0.329
Use of multivitamin					0.0250	
No	5.17	4.98; 5.36	(reference)			
Yes	5.81	5.03; 6.58	0.63	0.08; 1.19		0.023 *
Home-made meal frequency					0.0007	
Less than 3 days	5.18	4.68; 5.68	(reference)			
Greater than or equal to 3 days	5.29	5.07; 5.51	0.11	−0.47; 0.69		0.38
Pre-gestational BMI ^1^					0.0094	
Eutrophy	5.38	5.04; 5.71	(reference)			
Low weight	5.03	4.22; 5.83	−0.35	−1.41; 0.71		0.516
overweight	5.07	4.76; 5.39	−0.30	−0.79; 0.17		0.214
Obesity	5.30	4.83; 5.76	−0.08	−0.66; 0.50		0.784
Previous Pregnancy					0.0056	
Primiparous	5.41	5.12; 5.71	(reference)			
Multiparous	5.18	4.92; 5.44	−0.22	−0.64; 0.18		0.281
Planned Pregnancy					0.0151	
No	5.14	4.92; 5.37	(reference)			
Yes	5.51	5.15; 5.87	0.36	−0.03; 0.77		0.075 *
Energy intake by quintiles of UPF ^2^ (% of total energy intake)					0.1316	
Q1	5.57	5.00; 6.14	(reference)			
Q2	5.97	5.59; 6.35	0.39	−0.17; 0.97		0.173
Q3	5.13	4.79; 5.47	−0.44	−1.01; 0.13		0.132
Q4	5.31	4.95; 5.67	−0.26	−0.84; 0.31		0.373
Q5	4.40	4.06; 4.73	−1.17	−1.75; −0.59		<0.001 **

^1^ BMI: Body Mass Index; ^2^ UPF: ultra-processed food; * *p* < 0.05; ** *p* < 0.001.

**Table 3 ijerph-20-06063-t003:** Simple linear regression models of variables associated with folate consumption (mg/1000 kcal/day) of pregnant women assisted in Primary Health Care. Federal District, Brazil, 2019–2021 (n = 217).

Explanatory Variables	Mean Folate Consumption (mg/1000 kcal/day)	CI 95%	Non-Standard β	CI 95%	*R-Squared*	*p*-Value
Age					0.0087	
20 to 24 years	184.58	164.78; 204.38	(reference)			
25 to 29 years	196.84	171.67; 222.02	12.26	−17.46; 41.98		0.417
30 to 34 years	204.48	181.93; 227.04	19.90	−10.64; 50.44		0.200
≥35 years	189.45	167.58; 211.32	4.86	−28.66; 38.39		0.775
Gestational Period					0.0171	
1st Trimester	169.13	139.65; 198;60	(reference)			
2nd Trimester	202.29	183.71; 220.87	33.16	−1.30; 67.62		0.059
3rd Trimester	192.72	176.46; 208.97	23.59	−10.42; 57.60		0.173
Access to social welfare program					0.0064	
No	195.69	183.96; 207.42	(reference)			
Yes	174.69	132.18; 217.2	−20.99	−56.88; 14.88		0.250
Schooling					0.0185	
Elementary School	168.57	139.71; 197.43	(reference)			
High school	201.22	184.81; 217.64	32.65	−0.28; 65.59		0.052
University education	190.69	172.48; 208.89	22.11	−13.49; 57.73		0.222 *
Self-reported skin color					0.0081	
White	189.32	164.04; 214.60	(reference)			
black/brown	195.43	182.05; 208.82	6.11	−22.63; 34.86		34.86
Asian	157.01	120.86; 193.15	−32.31	−96.18; 31.56		31.56
Folic acid supplementation					0.0123	
No	184.65	170.01; 199.29	(reference)			
Yes	203.30	185.39; 221.21	18.64	−4.27; 41.57		0.110
Use of multivitamin					0.0005	
No	193.91	181.14; 206.68	(reference)			
Yes	188.43	161.04; 215.82	−5.48	−37.95; 26.98		0.739
Home-made meal frequency					0.0114	
Less than 3 days	170.25	144.78; 195.73	(reference)			
Greater than or equal to 3 days	196.51	183.90; 209.126	26.25	−7.27; 59.79		0.124
Pre-gestational BMI ^1^					0.0103	
Eutrophy	197.38	180.72; 214.05	(reference)			
Low weight	158.55	110.78; 206.23	−38.83	−99.42; 21.75		0.208
overweight	187.42	164.68; 210.15	−9.96	−37.69; 17.75		0.479
Obesity	197.12	168.07; 226.18	−0.26	−33.51; 32.99		0.988
Previous Pregnancy					0.0000	
No	193.86	175.78; 211;94	(reference)			
Yes	192.91	178.14; 207.69	−0.94	−25.14; 23.25		0.939
Planned Pregnancy					0.0186	
No	184.27	170.58; 197.96	(reference)			
Yes	207.73	188.104; 227.44	23.50	0.17; 46.82		0.048 *
Energy intake by quintiles of UPF ^2^ (% of total energy intake)					0.0788	
Q1	210.99	184.64; 237.35	(reference)			
Q2	217.47	189.69; 245.24	6.47	−27.56; 40.50		0.708
Q3	201.20	179.19; 223.22	−9.79	−44.02; 24.43		0.573
Q4	183.10	158.54; 207.66	−27.89	−62.33; 6.53		0.112
Q5	152.57	133.37; 171.78	−58.42	−92.85; −23.98		0.001 *

^1^ BMI: Body Mass Index; ^2^ UPF: ultra-processed food; * *p* < 0.05.

**Table 4 ijerph-20-06063-t004:** Final multiple linear regression model of variables associated with iron consumption in pregnant women assisted in Primary Health Care. Federal District, Brazil, 2019–2021 (n = 217).

Explanatory Variables	Non-Standard β	CI 95%	Standardized β	*p*-Value
Schooling				
Elementary School	(reference)			
High school	0.74	0.20; 1.28	0.25	0.007 *
University education	0.40	−0.20; 1.02	0.12	0.191
Energy intake by quintiles of UPF (% of total energy intake)				
Q1	(reference)			
Q2	0.37	−0.20; 0.95	0.10	0.203
Q3	−0.30	−0.89; 0.28	−0.08	0.314
Q4	−0.22	−0.81; 0.37	−0.06	0.461
Q5	−1.15	−1.74; 0.55	−0.31	<0.001 **

Note: Model adjusted by the variables Use of Ferrous Sulfate and Use of multivitamin. Backward Method. R^2^: 0.17. Adjusted R^2^: 0.14. Test F: *p* < 0.0001. * *p* < 0.05; ** *p* < 0.001.

**Table 5 ijerph-20-06063-t005:** Final multiple linear regression model of variables associated with folate consumption in pregnant women assisted in Primary Health Care. Federal District, Brazil, 2019–2021 (n = 217).

Explanatory Variables	Non-Standard β	CI 95%	Standardized β	*p*-Value
Schooling				
Elementary School	(reference)			
High school	38.95	6.96; 70.95	0.23	0.017 *
University education	24.88	−11.48; 61.26	0.13	0.179
Gestational Period				
1st Trimester	(reference)			
2nd Trimester	39.44	5.58; 73.30	0.22	0.023 *
3rd Trimester	24.87	−8.47; 58.23	0.14	0.143
Planned Pregnancy				
No	(reference)			
Yes	26.88	3.58; 50.18	0.15	0.024 *
Energy intake by quintiles of UPF (% of total energy intake)				
Q1	(reference)			
Q2	4.32	−30.19; 38.85	0.02	0.805
Q3	−13.45	−48.85; 21.94	−0.06	0.454
Q4	−37.69	−72.91; −2.47	−0.17	0.036 *
Q5	−63.23	−98.32; −28.15	−0.29	<0.001 **

Note: Model adjusted by the variables Use of Folic Acid and Use of Multivitamin. Backward Method. R^2^: 0.16. Adjusted R^2^: 0.11. Test F: *p* < 0.0001. * *p* < 0.05; ** *p* < 0.001.

## Data Availability

Data are available on reasonable request. The dataset used to conduct the analyses is available from the corresponding author on reasonable request.

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
