# Peer review of "Ultra-Processed Foods and Schooling Are Independently Associated with Lower Iron and Folate Consumption by Pregnant Women Followed in Primary Health Care"

_ijerph, 2023, doi:10.3390/ijerph20126063_

Round 1

Reviewer 1 Report

The manuscript entitled Ultra-processed foods and schooling are independently associated with lower Iron and Folate Consumption by Pregnant Women Followed in Primary Health Care is an attempt to process the data pertaining to intakes of micronutrients during the gestation period. However, the quality of manuscript can be further strengthened by addressing the following queries:

1.      Authors should avoid abbreviation in the abstract section.

2.      In the introduction section, research is focused on the Federal District of Brazil, authors should provide the world data of the problems associated with poor intake of iron and folate that renders morality to the fetus as well as mother.

3.      Materials and methods section, Details are not up to the mark. Authors should avoid the discussion of other projects focusing on other micronutrients. The details of iron and folate strategy need to be discussed for bringing the clarity of this present study.

4.      Table 1, have the authors used the data of the same pregnant women for all the trimesters of gestation period.

5.      Line 276, repetition, e.g. 49.2%, may be removed.

6.      Table 1-3, authors should discuss the trends of increasing/decreasing order.

7.      References should be uniform.

8.      The findings of the present data processing may be highlighted at the abstract section as well as the conclusion section.

Author Response

Comments and Suggestions for Authors

The manuscript entitled Ultra-processed foods and schooling are independently associated with lower Iron and Folate Consumption by Pregnant Women Followed in Primary Health Care is an attempt to process the data pertaining to intakes of micronutrients during the gestation period. However, the quality of manuscript can be further strengthened by addressing the following queries:

Answer: Dear Reviewer, we appreciate your thoughtful insights to improve the quality of our manuscript. Please find below point-by-point replies to the responses.

  1. Authors should avoid abbreviation in the abstract section.

Answer: Abbreviations have been removed from the abstract as recommended. Please see lines 14-36.

  1. In the introduction section, research is focused on the Federal District of Brazil, authors should provide the world data of the problems associated with poor intake of iron and folate that renders morality to the fetus as well as mother.

Answer: We have added the global and Brazil prevalence of anemia in pregnant women (lines 48-51) and information on the consequences and risk factors of folate deficiency during pregnancy (Lines 61-65).

  1. Materials and methods section, Details are not up to the mark. Authors should avoid the discussion of other projects focusing on other micronutrients. The details of iron and folate strategy need to be discussed for bringing the clarity of this present study.

Answer: The EMDI is a multi-center study proposed to evaluate iodine deficiency in pregnant women in Brazil and other variables such as socioeconomic, anthropometric, clinical, and dietary. Thus, all studies using data collected from EMDI must cite the major study according to the principal investigator in charge. The details about the evaluation of iron and folate intake methodology are described in lines 156-171.

  1. Table 1, have the authors used the data of the same pregnant women for all the trimesters of gestation period.

Answer: We would like to clarify that the study sample included women from all gestational trimesters. We better describe this information in line 119-120.

  1. Line 276, repetition, e.g. 49.2%, may be removed.

Answer: Thank you for your comment. We corrected the information in line 332.

  1. Table 1-3, authors should discuss the trends of increasing/decreasing order.

Answer: The data in Tables 1-3 were adjusted and presented in descending order in the results section. In the discussion section, we focus on the main result of the manuscript (Tables 4-5) in order to highlight our main findings because they answered our research question.

  1. References should be uniform.

Answer: The references section has been adjusted.

  1. The findings of the present data processing may be highlighted at the abstract section as well as the conclusion section.

Answer: Due to the limitation of words in the abstract, we informed only the main results in the abstract section (Please see lines 26-35). In the conclusion section, we rewrite our findings (Lines 427-429).

Reviewer 2 Report

The manuscript"Ultra-processed foods and schooling are independently associated with lower iron and folate comsumption by pregnant women followed in primary health care" is well-designed and interesting.  The paper could be published in IJERPH.  I only have some minor comments 

P15-17. The sentence was not clear.

P249 what dose during the first one thousand days of life mean?

P252 "such as " should be deleted.

P388 Another limitation was not to consider .....

Author Response

Comments and Suggestions for Authors

The manuscript"Ultra-processed foods and schooling are independently associated with lower iron and folate comsumption by pregnant women followed in primary health care" is well-designed and interesting.  The paper could be published in IJERPH.  I only have some minor comments

Answer: Dear Reviewer, we appreciate your thoughtful insights to improve the quality of our manuscript. Please find below point-by-point replies to the responses.

P15-17. The sentence was not clear.

Answer: The objective has been adjusted in lines 16-18.

P249 what dose during the first one thousand days of life mean?

Answer: Regarding the first days of the baby's life, we clarify that it is the 270 days of pregnancy and 730 days (from birth to 2 years of life). Brazil has health public policies that prioritize this period of the child's life. We rewrite this sentence, please see line 275.

P252 "such as " should be deleted.

Answer: Some words have been adjusted in the discussion as recommended.

P388 Another limitation was not to consider .....

Answer: We rewrite the sentence. Please see lines 415-416.

Reviewer 3 Report

In this MS, the authors tend to analyze the dietary, sociodemographic, and lifestyle risk factors associated with low iron and folate intake by pregnant women followed up in PHC in Brazil using a cross-sectional observational study. In general, this study is well organized and written. Thus, I can recommend this MS for publication in IJERPH after the following concerns are solved. My comments are as follows:

Major concerns:

1.      The authors declared their study included 217 pregnant women. In Table 1, the total number of participants in the variable section of “Access to social welfare program”, “Schooling”, “Self-reported skin color”, “Household Income (USD)”, “Ferrous sulfate supplementation”, “Folic acid supplementation”, “Use of multivitamins”, “Home-made meal frequency”, “Previous Pregnancy” and “Planned Pregnancy” were not consisted with N=217. Are they statistical errors? It is hard to understand because some of them are two classified variables. Please clarify.

2.      The authors should provide flow chart to show some details of the Inclusion and Exclusion Criteria. For example, “women of different gestational ages followed up at PHC were included in the study. Pregnant who had any physical changes that could distort the research results, such as a history of thyroid disease or surgery, a reported diagnosis of hypo or hyperthyroidism, were excluded from the study”, please be a bit more accurate to specific number.

3.      P<0.05 or P<0.001 in Table 2-5 should be indexed and add related descriptions in corresponding NOTE below your table.

4.      Some of the small paragraphs (1-2 sentences) should be merged to adjacent paragraphs. For example, Line 111-112 describes almost the same content with Line 113-125. In addition, the authors should re-organize your whole MS because too many small paragraphs in a paper were in inconformity to academic norm.

Minor concerns:

1.      Line 31, delete the redundant full stop.

2.      Line 225: R2, not R2.

3.      “P<0.05 or P<0.001”, P in the whole text should be in italics.

Author Response

In this MS, the authors tend to analyze the dietary, sociodemographic, and lifestyle risk factors associated with low iron and folate intake by pregnant women followed up in PHC in Brazil using a cross-sectional observational study. In general, this study is well organized and written. Thus, I can recommend this MS for publication in IJERPH after the following concerns are solved. My comments are as follows:

Answer: Dear Reviewer, we appreciate your thoughtful insights to improve the quality of our manuscript. Please find below point-by-point replies to the responses.

Major concerns:

  1. The authors declared their study included 217 pregnant women. In Table 1, the total number of participants in the variable section of “Access to social welfare program”, “Schooling”, “Self-reported skin color”, “Household Income (USD)”, “Ferrous sulfate supplementation”, “Folic acid supplementation”, “Use of multivitamins”, “Home-made meal frequency”, “Previous Pregnancy” and “Planned Pregnancy” were not consisted with N=217. Are they statistical errors? It is hard to understand because some of them are two classified variables. Please clarify.

Answer: Regarding the sample of 217 pregnant women, we clarify that some variables, such as “Access to social welfare program”, “Schooling”, “Self-reported skin color”, “Household Income (USD)”, “Ferrous sulfate supplementation”, “Folic acid supplementation”, “Use of multivitamins”, “Home-made meal frequency”, “Previous Pregnancy” and “Planned Pregnancy” presented missing data during the collection period. Also, some women refused to respond to these variables.

  1. The authors should provide flow chart to show some details of the Inclusion and Exclusion Criteria. For example, “women of different gestational ages followed up at PHC were included in the study. Pregnant who had any physical changes that could distort the research results, such as a history of thyroid disease or surgery, a reported diagnosis of hypo or hyperthyroidism, were excluded from the study”, please be a bit more accurate to specific number.

Answer: Thank you for your comment. We rewrite the Inclusion and Exclusion Criteria to better describe our sample (lines 119-126) and add the flowchart (Figure 1, line 201).

  1. P<0.05 or P<0.001 in Table 2-5 should be indexed and add related descriptions in corresponding NOTE below your table.

Answer: Thank you for your comment. The p values were adjusted in Table 2-5.

  1. Some of the small paragraphs (1-2 sentences) should be merged to adjacent paragraphs. For example, Line 111-112 describes almost the same content with Line 113-125. In addition, the authors should re-organize your whole MS because too many small paragraphs in a paper were in inconformity to academic norm.

Answer: Thank you for your comment. The small paragraphs have been adjusted, merging with underlying paragraphs in the manuscript as requested.

Minor concerns:

  1. Line 31, delete the redundant full stop.

  1. Line 225: R2, not R2.

  1. “P<0.05 or P<0.001”, P in the whole text should be in italics.

Answer: All the minor comments were corrected.
